# Metabolic network analysis reveals microbial community interactions in anammox granules

Christopher E. Lawson[1], Sha Wu[2], Ananda S. Bhattacharjee[2], Joshua J. Hamilton[3], Katherine D. McMahon[1,3], Ramesh Goel[2] & Daniel R. Noguera[1,4]

Microbial communities mediating anaerobic ammonium oxidation (anammox) represent one of the most energy-efficient environmental biotechnologies for nitrogen removal from wastewater. However, little is known about the functional role heterotrophic bacteria play in anammox granules. Here, we use genome-centric metagenomics to recover 17 draft genomes of anammox and heterotrophic bacteria from a laboratory-scale anammox bioreactor. We combine metabolic network reconstruction with metatranscriptomics to examine the gene expression of anammox and heterotrophic bacteria and to identify their potential interactions. We find that Chlorobi-affiliated bacteria may be highly active protein degraders, catabolizing extracellular peptides while recycling nitrate to nitrite. Other heterotrophs may also contribute to scavenging of detritus and peptides produced by anammox bacteria, and potentially use alternative electron donors, such as $H_2$, acetate and formate. Our findings improve the understanding of metabolic activities and interactions between anammox and heterotrophic bacteria and offer the first transcriptional insights on ecosystem function in anammox granules.

[1] Department of Civil and Environmental Engineering, University of Wisconsin–Madison, Madison, Wisconsin 53706, USA. [2] Department of Civil and Environmental Engineering, University of Utah, Salt Lake City, Utah 84112, USA. [3] Department of Bacteriology, University of Wisconsin–Madison, Madison, Wisconsin 53706, USA. [4] Great Lakes Bioenergy Research Center, Wisconsin Energy Institute, University of Wisconsin–Madison, Madison, Wisconsin 53726, USA. Correspondence and requests for materials should be addressed to R.G. (email: ram.goel@utah.edu) or to D.R.N. (email: dnoguera@wisc.edu).

Microbial communities mediating anaerobic ammonium oxidation (anammox) represent one of the most energy-efficient environmental biotechnologies for nitrogen removal from wastewater. The process is typically used to treat high strength ammonium wastewaters and offers significant cost savings compared to conventional nitrogen removal processes that require energy-intensive aeration for nitrification and also consume large quantities of organic carbon during denitrification[1]. In practice, anammox-based wastewater treatment systems are combined with a nitritation step in either a single-stage[2] or two-stage bioreactor system[3]. In these engineered ecosystems, a fraction of the ammonium is first oxidized to nitrite by aerobic ammonia oxidizing bacteria. Subsequently, anammox bacteria anaerobically oxidize the remaining ammonium directly to nitrogen gas using the produced nitrite as a terminal electron acceptor[4,5].

Five genera of anammox bacteria have been discovered to date, including *Kuenenia*, *Brocadia*, *Anammoxoglobus* and *Jettenia* commonly found in activated sludge, and *Scalindua* commonly found in marine environments[6]. These lineages all have 'Candidatus' status as they do not exist in pure culture and must be grown in laboratory enrichments. Because anammox bacteria have a slow growth rate[7], either biofilm reactors that use carrier media or granular sludge reactors are used to retain sufficient biomass in the system[8]. Such reactor configurations support the formation of dense microbial communities that can be readily separated from the liquid wastewater and enriched in the bioreactor. Anammox granules consist of a mixture of cell aggregates and abiotic particles embedded within a matrix of organic extracellular polymeric substances (EPS)[9–11]. EPS present in anammox granules has been found to contain high amounts of protein and polysaccharides, where increased hydrophobic amino acid content has been observed to be a main factor determining granule aggregation ability[11]. The high aggregate density also serves to limit oxygen diffusion into the granule interior, allowing for the proliferation of anammox and other anaerobic bacteria.

While most studies have focused on understanding the ecophysiology of anammox bacteria[4,6,12] and their interactions with autotrophic nitrifying bacteria[13], little is known about the activity of heterotrophic bacteria in anammox bioreactors. Previous studies based on 16S ribosomal RNA (rRNA) gene clone libraries and amplicon sequencing have shown that heterotrophic bacteria affiliated with the phyla Chlorobi, Bacteroidetes, Chloroflexi and Proteobacteria comprise a large fraction of the microbial community in anammox bioreactors[10,14,15]. Despite differences in bioreactor influent composition, these heterotrophic bacteria appear to share high phylogenetic similarity across different anammox wastewater treatment systems[16], suggesting that a more universal interaction exists between them and anammox bacteria.

The exact role of heterotrophic bacteria in anammox systems has not yet been determined, though a few clues have been uncovered. A recent metagenomic study revealed that most of the heterotrophic organisms in anammox granules encode the ability to respire nitrate via partial denitrification, possibly completing a nitrite loop with anammox and nitrite oxidizing bacteria (NOB) by reducing nitrate back to nitrite[16]. This activity could contribute to the removal of excess nitrate produced from the system during anammox growth or nitrite oxidation by NOB. Hydrolysis of EPS into soluble compounds and/or the secretion of soluble microbial products by anammox bacteria during cell growth is believed to support denitrification and heterotrophic growth, particularly for anammox bioreactors that receive no external organic carbon substrates[17,18]. However, the specific metabolite exchange reactions promoting interactions between heterotrophic and anammox bacteria remain poorly understood.

Here, we combine metagenomic and metatranscriptomic data to examine the gene expression of anammox and heterotrophic bacteria in a laboratory-scale anammox bioreactor and to identify their potential interactions. Metagenomic binning was used to recover near-complete population genomes from members of the microbial community inhabiting anammox granules. Resulting population genomes were used to reconstruct each organism's metabolic network and served as reference platforms for the profiling of gene expression at the community scale. We find that Chlorobi-affiliated bacteria may be highly active protein degraders, catabolizing extracellular peptides bound in the EPS matrix, while respiring nitrate, produced during anammox bacterial growth, to nitrite. Other heterotrophic bacteria may also contribute to the scavenging of detritus and peptides produced by anammox bacteria, and potentially use alternative electron donors, such as $H_2$, acetate and formate to fuel their energy metabolism. These findings improve the understanding of major metabolic activities and interactions occurring between anammox and heterotrophic bacteria and offer the first transcriptional insights on ecosystem function in anammox granules.

## Results

**Metagenomic sequencing and binning.** Sequencing of whole community DNA extracted from two separate biomass samples yielded a total of 41,063,036 reads after quality filtering (Supplementary Table 1). Co-assembly of the resulting reads using CLC Genomics Workbench generated a total of 83,770 contigs with an N50 of 2,376 bp and an N20 of 20,848 bp, accounting for 96% of the quality filtered DNA reads obtained from the second sampling event (Supplementary Table 2). Approximately 74% of the quality filtered mRNA reads could be mapped to this assembly, indicating that it also captured a large proportion of the metatranscriptome. Contigs were subsequently binned into population genomes based on tetranucleotide frequency and differential coverage using MetaBAT. This resulted in the recovery of 17 draft metagenome-assembled genomes (MAGs) affiliated with the phyla Planctomycetes, Chlorobi, Bacteroidetes, Chloroflexi, Proteobacteria and the candidate phylum Microgenomates (OP11; Table 1). Together, these genomes accounted for ∼82% and 59% of the total quality filtered DNA and mRNA reads obtained from the biomass samples, respectively, and therefore represented a major fraction of the microbial community present in the granular sludge reactor (Supplementary Table 2).

Figure 1 shows a phylogenetic tree of the recovered population genomes based on the protein sequences of 37 conserved bacterial marker genes. Many of the recovered genomes, particularly CHB1 and CFX5, had high similarity to genomes recently recovered from a single-stage reactor that were shown to share high 16S rRNA gene similarity to organisms detected in other anammox systems[16]. Surprisingly, the Chlorobi CHB1 genome was virtually identical to the Chlorobi OLB4 genome recovered by Speth *et al.*[16], sharing an average nucleotide identity of 99.8% despite the significant differences in reactor operation (two-stage versus single-stage), wastewater influent composition (potato-processing wastewater versus anaerobic digester filtrate) and geographical location (USA versus Netherlands; Supplementary Data 1).

**Microbial community abundance and gene expression.** We used the reads per kilobase per million mapped reads (RPKM) values for metagenomic reads and transcripts that mapped to each MAG as proxies for relative abundance and gene expression, respectively (Fig. 2). A summary of the metatranscriptomic sequencing statistics and read mapping can be found in

**Table 1 | Genome statistics of MAGs recovered from the anammox community.**

| Bin ID | Locus tag | Taxonomy | Completeness (%) | Contamination (%) | Genome size (bp) | # Scaffolds | N50 (scaffolds) | GC (%) | Predicted genes |
|---|---|---|---|---|---|---|---|---|---|
| AMX1 | UTAMX1 | Bacteria; Planctomycetes; Planctomycetia; Planctomycetales; Planctomycetaceae; *Brocadia* | 96 | 1 | 3,142,066 | 36 | 137,788 | 42.3 | 2,724 |
| AMX2 | UTAMX2 | Bacteria; Planctomycetes; Planctomycetia; Planctomycetales; Planctomycetaceae; *Brocadia* | 96 | 2 | 3,437,337 | 148 | 32,519 | 45.1 | 3,064 |
| PLA1 | UTPLA1 | Bacteria; Planctomycetes | 82 | 2 | 4,435,946 | 642 | 8,518 | 60.8 | 4,062 |
| BCD1 | UTBCD1 | Bacteria; Bacteroidetes; Sphingobacteriia; Sphingobacteriales | 96 | 1 | 3,757,364 | 58 | 91,406 | 41.0 | 3,180 |
| CFX1 | UTCFX1 | Bacteria; Chloroflexi; Anaerolineae | 91 | 2 | 2,777,515 | 34 | 125,597 | 52.6 | 2,576 |
| CFX2 | UTCFX2 | Bacteria; Chloroflexi; Anaerolineae | 81 | 5 | 2,924,761 | 111 | 48,865 | 57.0 | 2,622 |
| CFX3 | UTCFX3 | Bacteria; Chloroflexi; Anaerolineae | 82 | 1 | 2,655,948 | 281 | 12,640 | 61.3 | 2,676 |
| CFX4 | UTCFX4 | Bacteria; Chloroflexi | 81 | 5 | 4,654,268 | 672 | 8,164 | 54.8 | 4,526 |
| CFX5 | UTCFX5 | Bacteria; Chloroflexi; Anaerolineae | 84 | 2 | 3,776,973 | 364 | 12,671 | 62.8 | 3,432 |
| CHB1 | UTCHB1 | Bacteria; Chlorobi; Ignavibacteria; Ignavibacteriales; Melioribacteraceae | 95 | 0 | 2,424,598 | 19 | 211,700 | 37.6 | 2,083 |
| CHB2 | UTCHB2 | Bacteria; Chlorobi; Ignavibacteria; Ignavibacteriales; Ignavibacteriaceae | 95 | 0 | 4,062,427 | 225 | 36,474 | 33.0 | 3,556 |
| CHB3 | UTCHB3 | Bacteria; Chlorobi; Ignavibacteria; Ignavibacteriales | 83 | 2 | 2,943,821 | 349 | 10,790 | 42.1 | 2,526 |
| CHB4* | UTCHB4 | Bacteria; Chlorobi; Ignavibacteria; Ignavibacteriales | 43 | 0 | 1,189,945 | 296 | 4,153 | 34.7 | 1,311 |
| CHB5* | UTCHB5 | Bacteria; Chlorobi; Ignavibacteria; Ignavibacteriales | 6 | 2 | 1,684,758 | 362 | 4,608 | 34.4 | 1,621 |
| PRO1 | UTPRO1 | Bacteria; Proteobacteria; Deltaproteobacteria; Myxococcales; Polyangiaceae | 74 | 2 | 3,818,019 | 45 | 114,904 | 69.8 | 3,321 |
| PRO2 | UTPRO2 | Bacteria; Proteobacteria; Betaproteobacteria; Rhodocyclales; Rhodocyclaceae; Sulfuritalea | 68 | 1 | 2,355,954 | 380 | 7,446 | 66.9 | 2,730 |
| CPR1 | UTCPR1 | Bacteria; Microgenomates | 68 | 0 | 892,243 | 91 | 13,172 | 37.7 | 959 |

*indicates incomplete MAGs not submitted to GenBank.
GenBank accession numbers for each MAG can be found in Supplementary Table 3. GC, guanine-cytosine content.

Supplementary Table 2. Overall, gene expression corresponded with abundance in the anammox granules. Genomes affiliated with *Brocadia* (AMX1) and Chlorobi (CHB1) dominated the abundance and gene expression of the microbial community in the anammox granules (Fig. 2). AMX1 had a relative abundance and gene expression of ∼62 and 54%, respectively, whereas CHB1 had a relative abundance and gene expression of ∼21 and 26%, respectively. Other organisms that displayed moderate abundance and gene expression in the anammox ecosystem were affiliated with the phyla Chlorobi, Bacteroidetes, Chloroflexi and Proteobacteria (Fig. 2). Interestingly, several low abundance genomes were also observed to be particularly active based on gene expression (that is, PRO1, PRO2 and PLA1), as has been previously observed in other activated sludge ecosystems[19].

**Metabolic reconstruction of microbial community**. To examine the functional potential and gene expression of the anammox community, open reading frames (ORFs) were predicted and annotated across each MAG. Subsequently, mRNA transcripts were mapped against all ORFs to identify functions that were highly expressed by the community during steady-state bioreactor operation (Supplementary Data 2). Metabolic pathways were then reconstructed for each of eight near-complete MAGs that displayed high abundance and/or gene expression in the community (>0.5%) using MetaPathways 2.5 (ref. 20). A complete list of the inferred pathways and associated enzymes of each MAG can be found in Supplementary Data 3.

**Anammox metatranscriptomic insights**. As expected, genes involved in anammox metabolism from AMX1 were among the highest genes expressed in the community (Supplementary Data 2). Most notably, UTAMX1_1243, UTAMX1_1246 and UTAMX1_1249 annotated as hydrazine dehydrogenase (*hdh*), hydroxylamine oxidoreductase (*hao*) and hydrazine synthase subunit A (*hzsA*) were highly expressed, maintaining 10–20-fold coverage above median gene expression levels in the AMX1 genome (Supplementary Fig. 1). Six other *hao*-like genes were also identified in the AMX1 genome and displayed above median gene expression levels. Reciprocal best-BLAST searches confirmed that these genes were orthologous to *hao*-like genes from publically available anammox genomes (Supplementary Fig. 1). Among the Hao-like proteins, UTAMX1_1246 had the highest gene expression, suggesting it played an important role in anammox metabolism. This gene is orthologous to previously characterized Hao proteins from 'Candidatus Kueneia stuttgartiensis' (kustc1061)[5,21], 'Candidatus Brocadia anammoxidans'[22] and 'Candidatus Jettenia caeni'[23], which have been shown to oxidize hydroxylamine to nitric oxide and were also observed to be among the most abundant proteins expressed in these organisms. Phylogenetic analysis showed that AMX1 was closely related to both 'Candidatus Brocadia sinica'[24] and 'Candidatus Brocadia fulgida'[25] (Fig. 1). Similar to 'Ca. Brocadia sinica', neither *nirK* nor *nirS* were found in the AMX1 genome. This is consistent with the recently proposed hydroxylamine-dependent anammox mechanism in 'Ca. Brocadia sinica' that first reduces nitrite to hydroxylamine (instead of nitric oxide), and subsequently converts

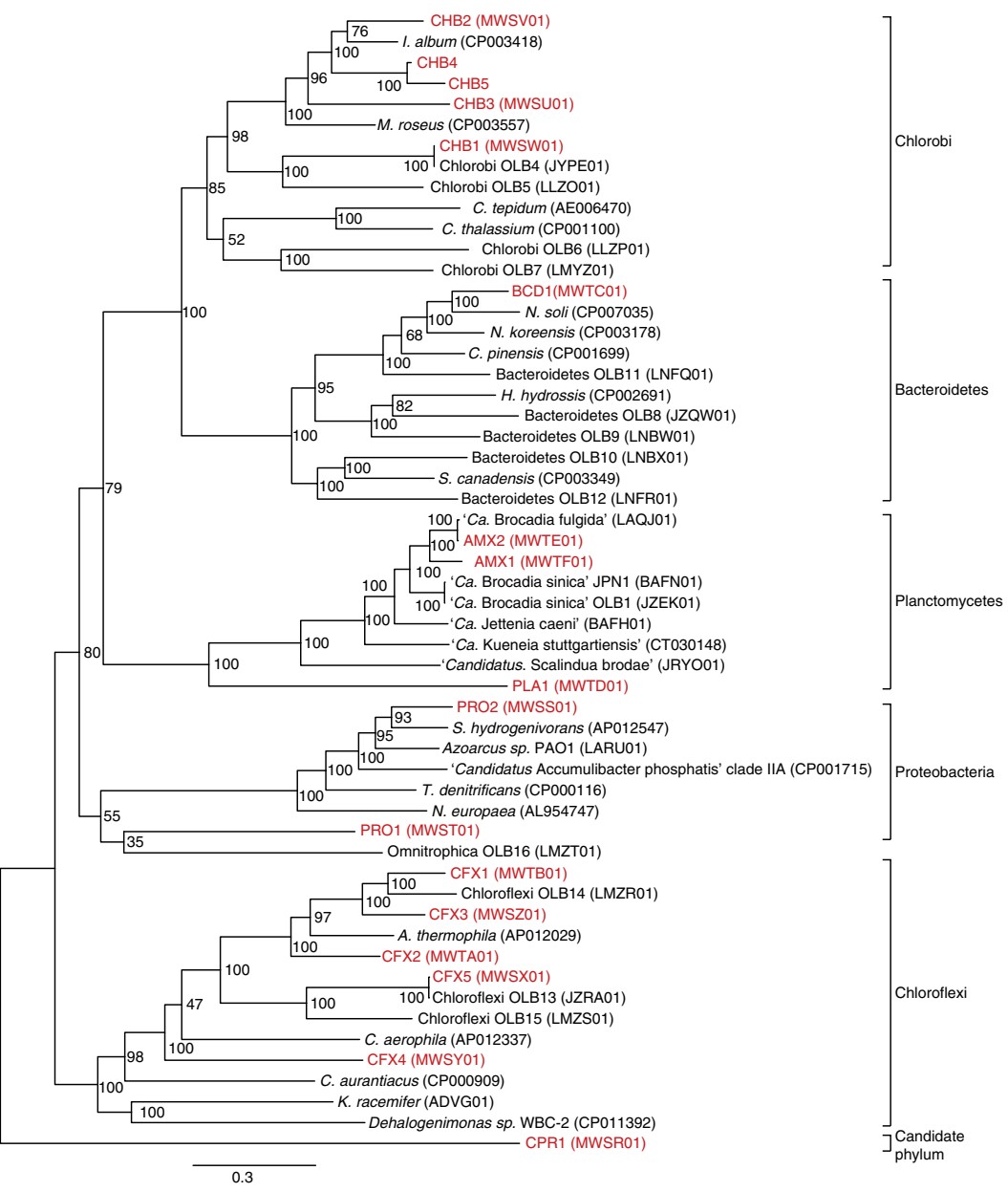

**Figure 1 | Phylogenetic tree of all recovered draft genomes from the anammox bioreactor.** Tree includes MAGs recovered from this study (red) and closely related genomes downloaded from the NCBI genome repository. GenBank accession numbers for each genome are provided in parentheses. Branch node numbers represent bootstrap support values. The tree was constructed using RAxML based on a set of 37 concatenated universal single-copy marker genes.

hydroxylamine and ammonium to hydrazine[26] (Supplementary Fig. 1). It has been proposed that Hao-like proteins lacking a crosslinking tyrosine in the c-terminus may be involved in the initial nitrite reduction reaction to either nitric oxide[5] or hydroxylamine[26], for which an enzyme has not yet been identified. The *hao*-like genes UTAMX1_1996, UTAMX1_1759 and UTAMX1_1192 expressed in the AMX1 genome that lack a crosslinking tyrosine might fulfil this role (Supplementary Fig. 1).

**Denitrification gene expression in anammox granules.** All heterotrophic organisms in the anammox granules encoded capabilities for partial or full denitrification (Fig. 3). This was similar to a recent report from a single-stage reactor by Speth *et al.*[16] Genes involved in denitrification were highly expressed in

the genomes, consistent with oxidative phosphorylation coupled to nitrate respiration being a dominant form of heterotrophic energy metabolism in the anammox granules (Supplementary Data 2 and 4). All heterotrophs expressed respiratory nitrate reductase genes (*narGHIJ*) that reduce nitrate to nitrite and require the transport of nitrate into the cytoplasm (Fig. 3; Supplementary Data 4). However, it should be noted that the heme-containing membrane anchor subunit (*narI*) gene had no detectable expression in CHB2 or CFX2, suggesting that this enzyme had low activity in these organisms. A periplasmic nitrate reductase (*napABCGH*) that does not contain a coupling site for proton motive force (PMF) generation was also expressed by PRO2. Pathways for dissimilatory nitrate reduction to ammonia via pentaheme nitrite reductase genes (*nrfHA*) were expressed by the Chlorobi genome CHB2 and the Chloroflexi genome CFX2 (Fig. 3; Supplementary Data 4). Chloroflexi (CFX1 and CFX2)

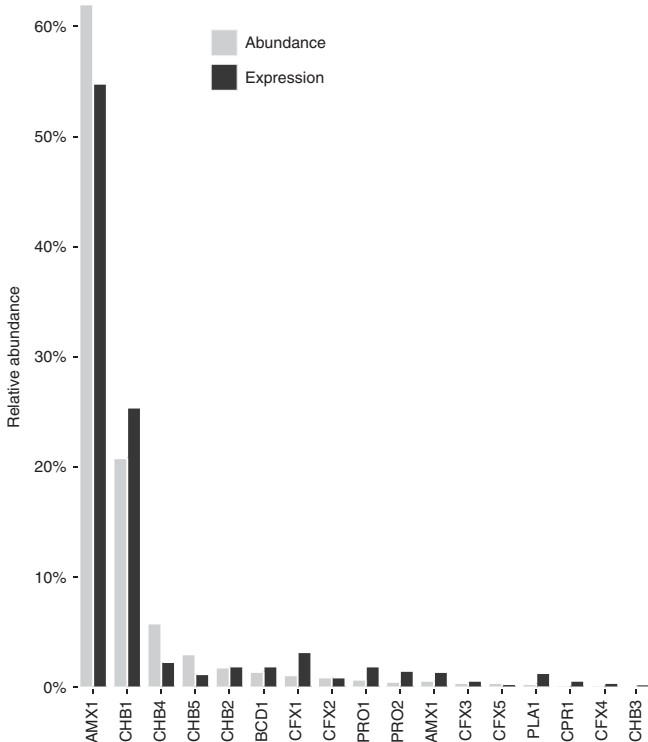

**Figure 2 | Abundance and gene expression of organisms represented by the MAGs recovered from the anammox bioreactor.** Abundance and gene expression estimates were based on RPKM values of metagenomic reads and transcripts that mapped to each MAG, respectively. See Supplementary Table 2 for mapping details.

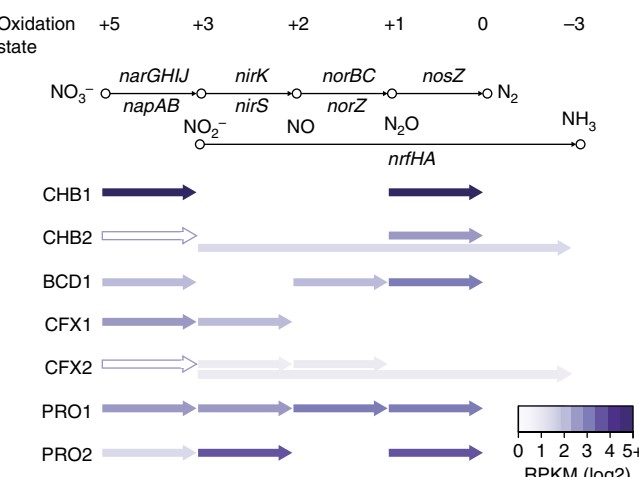

**Figure 3 | Presence and expression of denitrification genes across the recovered heterotrophic genomes.** Purple arrows indicate gene presence. Colour intensity represents gene expression (log2 RPKM), based on mapping of metatranscriptomic reads to the metagenomic assembly. An outlined white arrow indicates one or more enzyme subunits had no detectable gene expression. A summary of genes involved in denitrification across the recovered genomes can be found in Supplementary Data 4.

and Proteobacteria (PRO1 and PRO2) genomes also expressed genes for the reduction of nitrite to nitric oxide, either through a copper containing nitrite reductase (*nirK*) or cytochrome cd1 nitrite reductase (*nirS*; Fig. 3; Supplementary Data 4). The reduction of nitric oxide to nitrous oxide, either through the

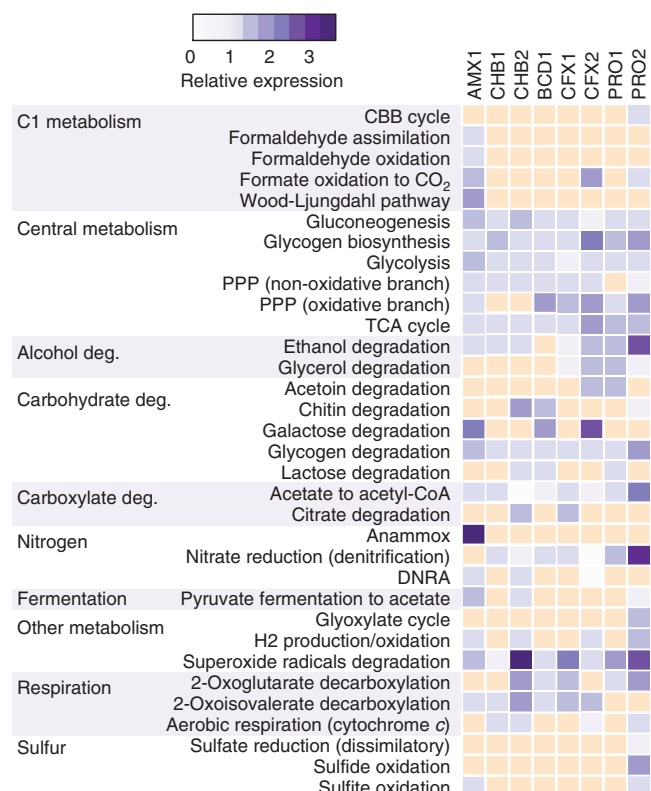

**Figure 4 | Relative gene expression of major carbon and energy metabolic pathways encoded by each MAG.** Colour intensity represents gene expression, based on mapping of metatranscriptomic reads to the metagenomic assembly. Gene expression was relativized by median RPKM values calculated across each ORF in a given MAG (Methods section). A value of 1 equals median expression in a given genome. Yellow box indicates pathway absence. Anammox metabolic pathway in the AMX1 genome had a relative expression value of 15. See Supplementary Data 3 for a detailed summary of all reconstructed metabolic pathways.

cytochrome *c*-dependent nitric oxide reductase (*norBC*) or the quinol-dependent nitric oxide reductase (*norZ*), was expressed by the BCD1, CFX2 and PRO1 genomes. Finally, Chlorobi (CHB1 and CHB2), Proteobacteria (PRO1 and PRO2), and the Bacteroidetes (BCD1) genomes expressed genes capable of reducing nitrous oxide to nitrogen gas via nitrous oxide reductase (*nosZ*). Together, these functions could facilitate a nitrite loop with anammox bacteria or support complete denitrification, thus enhancing overall nitrogen removal performance in the bioreactor.

**Carbon and energy metabolism of heterotrophic bacteria.** Figure 4 shows the gene expression profiles of major carbon and energy metabolic pathways across the anammox community. Only two low abundance heterotrophic organisms, BCD1 and PRO2, encoded known pathways for $CO_2$ fixation that would permit mixotrophic growth (Fig. 4; Supplementary Data 3). BCD1 displayed above median gene expression of PEP carboxylase (UTBCD1_1894), which allows $CO_2$ fixation into oxaloacetate (Supplementary Data 2); PRO2 displayed above median gene expression of Ribulose-1,5-bisphosphate carboxylase (RuBisCO, UTPRO2_2197), which would permit $CO_2$ fixation via the Calvin-Benson-Bassham cycle (Fig. 4; Supplementary Data 3). While all heterotrophs, except CHB1, expressed genes encoding pyruvate:flavodoxin/ferredoxin oxidoreductase (PFOR) and

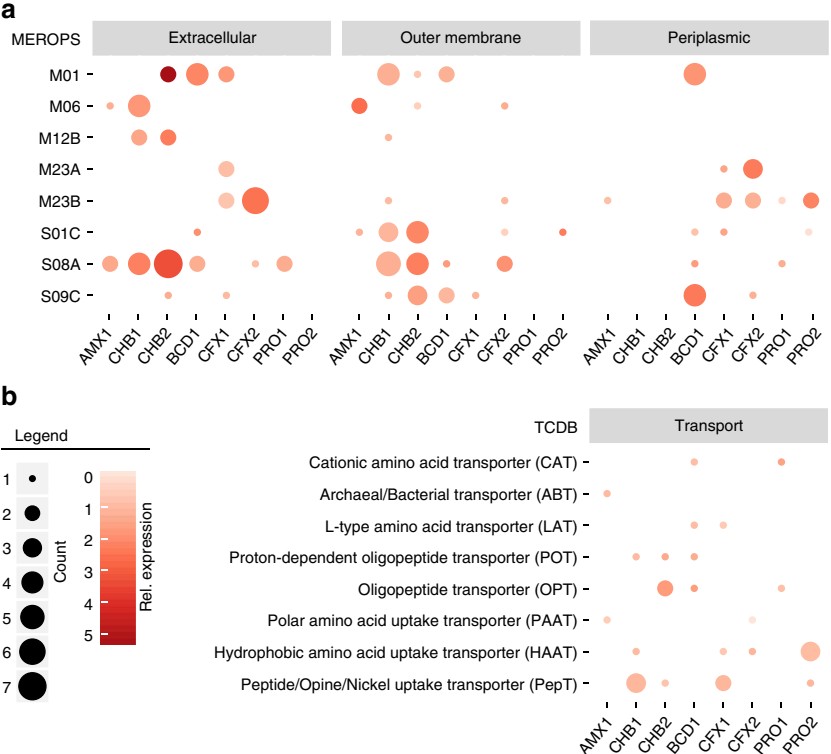

**Figure 5 | Predicted peptidases and amino acid transporters recovered from the MAGs.** (**a**) Number (bubble diameter) and relative gene expression (bubble colour intensity) of selected peptidases possibly involved in EPS matrix protein degradation. Peptidases were annotated against the MEROPS database[66]. The subcellular location (extracellular, outer membrane or peroplasm) of each peptidase was predicted using the subcellular localization predictor (CELLO)[69]. A summary of all predicted peptidases can be found in Supplementary Data 5. (**b**) Number (bubble diameter) and relative gene expression (bubble colour intensity) of amino acid and peptide transporters predicted across the recovered genomes. Transporters were annotated against the transporter classification database (TCDB)[70] and can be found in Supplementary Data 2.

2-oxoglutarate ferredoxin oxidoreductase (KFOR) that are commonly implicated in the reductive TCA cycle, other key genes of this pathway, such as ATP-citrate lyase and fumarate reductase, were missing from the genomes. As PFOR and KFOR are known to participate in other metabolic pathways (for example, KFOR may participate in the oxidative TCA cycle[27]), it is likely that these genes are not used for $CO_2$ fixation by heterotrophs in the community.

All organisms expressed genes involved in central carbon metabolic pathways, including glycolysis/gluconeogenesis, glycogen synthesis/degradation, the TCA cycle and the pentose phosphate pathway (PPP; Fig. 4; Supplementary Data 3). Surprisingly, the oxidative branch of the PPP was completely missing in both Chlorobi bacterium genomes (CHB1 and CHB2), suggesting that other enzymes are important for NADPH generation in these organisms. Genes for the oxidative PPP were also confirmed to be missing from the closely related Chlorobi bacterium OLB4 and OLB6 genomes[16], although they were present in the genomes of *Ignavibacterium album*[28] and *Melioribacter roseus*[29]. An alternative route for NADPH generation in CHB1 and CHB2 is likely through $NAD(P)^+$ transhydrogenase[30], which transfers electrons from NADH to $NADP^+$ and had above median gene expression in both genomes (Supplementary Data 2).

Genes encoding pathways for pyruvate fermentation to acetate and $CO_2$ via PFOR, phosphate acetyltransferase (*pta*) and acetate kinase (*ack*) were expressed in the CHB2 and PRO2 genomes (Fig. 4; Supplementary Data 3). While these genes could facilitate ATP synthesis via substrate level phosphorylation, it is also possible that the *pta-ack* reactions encoded by CHB2 and PRO2 are used for acetate consumption rather than production. These reactions form

a lower affinity pathway for acetate assimilation compared to the high-affinity acetyl-CoA synthetase (*acs*) pathway[31]. Under this scenario, however, only PRO2 could use acetate as a sole carbon source because it is the only genome that encodes the glyoxylate cycle (that is, malate synthase and isocitrate lyase). The AMP-forming *acs* gene was expressed in all genomes, whereas the ADP-forming *acs* gene was expressed in CHB1, CHB2, CFX1, CFX2 and PRO2 (Supplementary Data 2 and 3). Since these reactions are also reversible, with acetate formation resulting in ATP conservation, it is unclear whether the *acs* and/or *pta-ack* acetate pathways expressed by the microbial community are involved in acetate formation, acetate assimilation or both.

The CHB2 genome also expressed an eight-gene cluster (UTCHB2_2599 to UTCHB2_2606) ferredoxin:NAD+ oxidoreductase (*rnfCDGEAB*) complex and two putative electron-bifurcating hydrogenase gene clusters (*hydABC*; UTCHB2_709-711 and UTCHB2_1049-1051). The Rnf complex could couple the formation of a proton gradient with ATP synthesis by oxidizing reduced ferredoxin produced from the electron-bifurcating hydrogenase and/or PFOR[32]. This energy conservation mode was also identified in the closely related *I. album*[28] and *M. roseus*[29] genomes, however no orthologs were detected in the CHB1 genome.

The ability to use $H_2$, sulfite, formate, ethanol and other simple organic compounds as electron donors could also support the energy metabolism of the community (Fig. 4; Supplementary Data 3). CHB2, CFX2 and PRO2 expressed genes encoding a NAD-dependent hydrogenase that could be involved in $H_2$ oxidation. CFX2 also expressed genes encoding a formate hydrogenlyase complex (UTCFX2_2421-2429) that directly links formate oxidation to $H_2$ production[33]. Organisms encoding the

ability to use formate for energy conservation included CFX2 and PRO2 (Fig. 4; Supplementary Data 3). These organisms expressed genes encoding NAD-dependent formate dehydrogenase that catalyse the oxidation of formate to $CO_2$, donating electrons to NAD$+$ that could be used to generate a PMF. PRO2 also expressed genes encoding formate dehydrogenase O (FDH-O; UTPRO2_1096 – UTPRO2_1099) that would donate electrons to the quinone pool and could also be used for PMF generation[34]. Because no organism encoded the ability to produce formate via pyruvate formate lyase, it is possible that formate is made available to the community via $CO_2$ reduction to formate by AMX1, which is the first step in $CO_2$ fixation via the reductive acetyl-CoA pathway.

Unexpectedly, genes encoding aerobic respiration machinery were expressed by some community members, despite the lack of oxygen in the bioreactor. CHB1, CFX2 and PRO2 expressed genes encoding a low-affinity aa3-type terminal cytochrome $c$ oxidase; CHB2 expressed a high-affinity cytochrome $d$ ubiquinol oxidase gene (Fig. 4; Supplementary Data 3). While this could allow these organisms to respire oxygen under aerobic conditions, it is also possible that these oxidases have evolved for protection against oxygen[35], perform some unknown function, or are post-transcriptionally repressed. All organisms were also found to express genes encoding superoxide dismutase (Fig. 4; Supplementary Data 3), which may additionally function to protect against oxidative stress.

**Amino acid and carbohydrate catabolism.** The breakdown of EPS produced by anammox bacteria has been proposed to be a major organic carbon source supporting heterotrophic growth in anammox granules[11,17]. In agreement with this, genes encoding a wide range of peptidases were expressed in the heterotrophic genomes (Fig. 5; Supplementary Data 5). In particular, CHB1 and CHB2 encoded many extracellular subtilisin-like serine peptidases and metallopeptidases (Fig. 5; Supplementary Data 5). These genes were among the highest expressed ORFs in the genomes and may be involved with growth on proteinaceous substrates[36,37]. Consistent with this function, the CHB1 and CHB2 genomes expressed genes for the transport and catabolism of peptides and amino acids to central carbon intermediates (Figs 5 and 6; Supplementary Data 2 and 3). This would allow CHB1 and CHB2 to use amino acids as a carbon and energy source, in addition to direct assimilation into protein biosynthesis. Interestingly, ranking of the amino acids according to their biosynthetic cost[38] revealed that all organisms in the anammox granules lacked catabolic pathways for many amino acids with a high biosynthetic cost (Fig. 6; Supplementary Data 3). This may suggest that heterotrophic community members have been selected to preferentially degrade amino acids that are less costly to synthesize, while directly incorporating more costly amino acids into protein synthesis. Other heterotrophic genomes (BCD1, CFX1, CFX2, PRO2 and PRO1) also expressed genes involved in the transport and degradation of amino acids (Figs 5 and 6; Supplementary Data 2). These organisms could potentially take advantage of substrates made available by CHB1 and CHB2, or contribute to extracellular protein degradation as well.

Microbial community members also expressed diverse genes involved in the hydrolysis of carbohydrate bonds (Supplementary Data 6). In particular, CHB2 and BCD1 expressed many ORFs encoding glycoside hydrolases, some of which were predicted to be extracellular. The high number of these genes in the CHB2 genome appeared to be a major function differentiating it from the CHB1 genome, which had the lowest number of predicted glycoside hydrolyases in the community (Supplementary Data 6). CHB2 expressed several

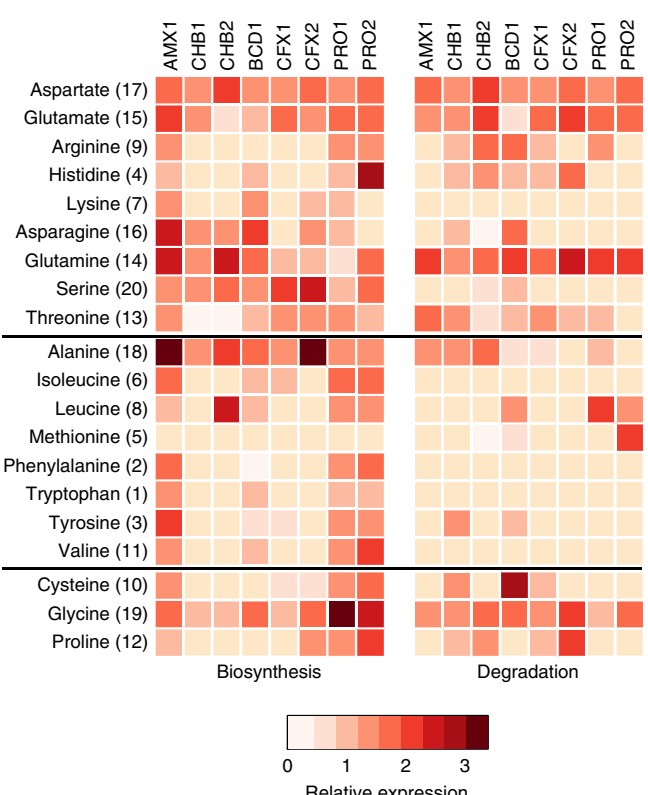

**Figure 6 | Relative gene expression of amino acid biosynthetic and degradation pathways encoded by each MAG.** Red colour intensity represents gene expression, which was relativized by median RPKM values calculated across each ORF in a given MAG. A value of 1 equals median expression. Top, middle and bottom panels separate amino acids by hydrophilic, hydrophobic and special structured side chains, respectively. Bracketed numbers rank the metabolic cost of amino acid biosynthesis based on values reported by Akashi and Gojobori[38], with 1 being the most costly.

genes that may act on glycosidic bonds found in polysaccharides, whereas BCD1 expressed a range of glycoside hydrolase genes that may use oligosaccharides, mucins and glycolipids as substrates and play an important role in carbohydrate breakdown for the community.

**Vitamin and amino acid auxotrophy.** Metabolite exchange of amino acids and vitamins is known to shape microbial community assembly[39]. We observed that several of the abundant heterotrophic bacteria, namely CHB1, CHB2, CFX1 and CFX2, were missing pathways for the synthesis of many hydrophobic amino acids (Fig. 6; Supplementary Data 3). This was in contrast to the AMX1 genomes and other heterotrophic genomes (BCD1, PRO1 and PRO2) that expressed pathways for the synthesis of most amino acids.

Several of the heterotrophic bacteria were also missing key genes involved in B-vitamin biosynthesis. The CHB1, CHB2 and CFX2 genomes were missing key genes involved in thiamin (vitamin B1) biosynthesis (thiamine-phosphate synthase and thiamine-monophosphate kinase), biotin (vitamin B7) biosynthesis (adenosylmethionine-8-amino-7-oxononanoate aminotransferase and bioin synthase) and adenosylcobalamin (vitamin B12) biosynthesis (cobalamin synthase and adenosylcobinamide-phosphate synthase; Supplementary Data 3). CFX1, BCD1, PRO2 and PRO1 also lacked key genes involved in

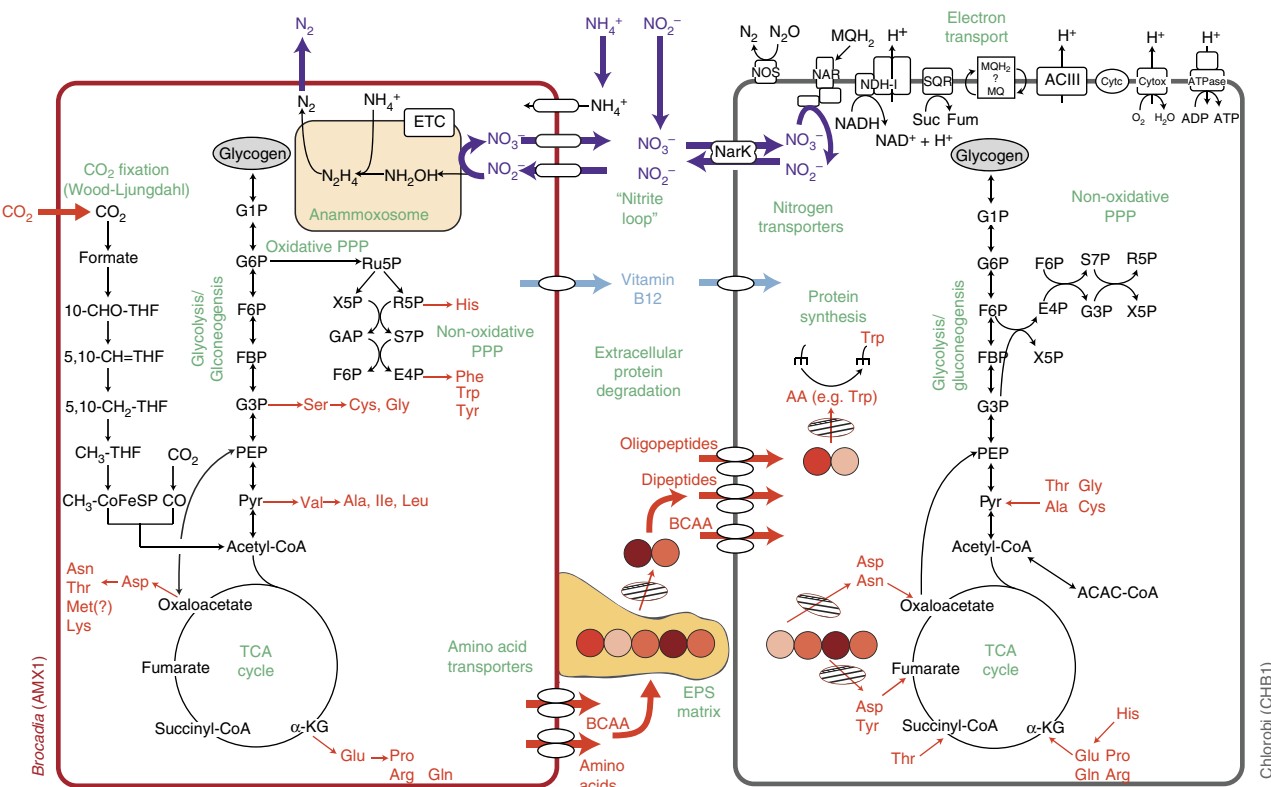

**Figure 7 | Proposed metabolic interactions between *Brocadia* (AMX1) and Chlorobi (CHB1) in anammox granules.** AMX1 fixes $CO_2$ and synthesizes amino acids and EPS. CHB1 degrades proteins bound in EPS using extracellular peptidases and subsequently transports and catabolizes short peptides (circles) to central carbon intermediates. Nitrite oxidation and reduction by AMX1 and CHB1, respectively, results in a distributed nitrite loop. Purple arrows indicate nitrogen cycling; orange arrows indicate carbon cycling; light blue arrows indicate vitamin B metabolite exchange. Hatched ovals indicate peptidases. BCAA, branched-chain amino acids. Met(?), methionine synthase present in genome but other steps involved in Met biosynthesis not identified. The presence of a periplasm has been ignored for clarity of the schematic.

adenosylcobalamin *de novo* synthesis. These genes were expressed in the AMX1 genome, suggesting that anammox bacteria may support B-vitamin requirements for the community. Genes encoding thiamin ABC-type transporters were expressed in the PRO1 and PRO2 genomes, whereas the CFX1, CFX2, PRO1 and PRO2 genomes expressed ABC-type transporters for vitamin B12 (*btuFCD*; Supplementary Data 2). The AMX1 (UTAMX1_2357), CHB1 (UTCHB1_731), CHB2 (UTCHB2_2713) and BCD1 (UTBCD1_3076) genomes expressed genes encoding the TonB-dependent vitamin B12 transporter (*btuB*), which facilitates vitamin B12 translocation across the outer membrane and may also be involved in uptake.

## Discussion

The integration of metagenomic and metatranscriptomic sequencing allowed us to examine gene expression and microbial interactions in a lab-scale anammox community at the ecosystem-scale. Our results revealed that anammox bacteria affiliated with *Brocadia* (AMX1) and heterotrophic bacteria affiliated with Chlorobi (CHB1) dominated the abundance and gene expression of the anammox granule community. Other less abundant heterotrophic bacteria affiliated with Chloroflexi, Bacteroidetes and Proteobacteria also displayed significant gene expression that may contribute to ecosystem function. The high similarity between these genomes and the genomes recently recovered from a single-stage anammox reactor[16] suggests that a core microbiome exists in anammox-based wastewater treatment systems, despite differences in reactor operation and influent wastewater composition.

Remarkably, the CHB1 genome was nearly identical to the Chlorobi bacterium OLB4 genome recovered by Speth *et al*[16] (99.8% average nucleotide identity) that achieved the second highest abundance in a full-scale partial-nitritation-anammox reactor from the Netherlands[16,40], suggesting that this organism interacts consistently with *Brocadia sp.* in anammox-based wastewater treatment systems. Figure 7 summarizes the major metabolic interactions proposed between AMX1 and CHB1 in anammox granules, based on the metabolic reconstruction and gene expression results obtained in this study. We posit that the ecological role of CHB1 in anammox granules is to degrade and catabolize extracellular peptides bound in the EPS matrix, while respiring nitrate produced during anammox bacterial growth to nitrite. This is supported by the high gene expression of respiratory nitrate reductase and extracellular peptidases in the CHB1 genome, as well as transporters and genes involved in amino acid catabolism. Interestingly, the CHB1 genome lacked catabolic pathways for amino acids that carry a high biosynthetic cost, suggesting that less costly amino acids (for example, glutamine) may be preferentially used as carbon and energy sources by heterotrophic bacteria in anammox granules. Such preferential degradation of amino acids has also been observed in anoxic water columns with sinking particulate organic carbon[41].

Because CHB1 lacks many pathways for amino acid biosynthesis, it is likely that peptide and amino acid substrates used as carbon and energy sources originate from *Brocadia sp.*, which derives carbon via $CO_2$ fixation and expressed pathways for the synthesis of nearly all amino acids (Fig. 6). This is consistent with [14]C-bicarbonate tracer experiments that showed heterotrophic bacteria in anammox bioreactors fed with no external organic

**Table 2 | Process performance data from the two-stage PNA bioreactor at steady-state operation.**

| | pH | COD (mg l$^{-1}$) | Ammonia (mgN l$^{-1}$) | Nitrite (mgN l$^{-1}$) | Nitrate (mgN l$^{-1}$) | VSS (mg l$^{-1}$) | Activity (kg NH$_4$-N kg$^{-1}$ VSS-day) |
|---|---|---|---|---|---|---|---|
| Filtrate | 8.1 | 535 ± 10.8 | 716.0 | 69.6 | 20.8 | NA | NA |
| PN-effluent | 6.8 | 405 ± 12.2 | 462.0 | 424.8 | 46.2 | NA | NA |
| Anammox-effluent | 6.9 | 350 ± 14.7 | 56.0 | 28.0 | 48.2 | NA | NA |
| Anammox reactor | 6.8–8.0 | NA | NA | NA | NA | 802 | 0.309 ± 0.0094 |
| PN reactor | 6.5–8.5 | NA | NA | NA | NA | 643 | 0.419 ± 0.0265 |

COD, chemical oxygen demand; PN, partial nitritation; VSS, volatile suspended solids.

carbon compounds degraded and utilized cellular components produced by anammox bacteria[42]. The high transcript abundance of CHB1 in the metatranscriptome (25% relative community expression) also suggests that proteolysis by this organism is a significant component of carbon flux in anammox granules, possibly making amino acids available to other microbial community members. The number and high gene expression of amino acid transporters across the other heterotrophic genomes supports this interaction, suggesting that protein degradation by CHB1 may be a significant contributor to community assembly in anammox granules, similar to particle degrading taxa in marine ecosystems[43].

It is also possible that *Brocadia sp.* supply B-vitamins to members of the microbial community, based on the expression of key genes involved in their biosynthesis and auxotrophies present across the heterotrophic genomes. The exchange of B-vitamins has been postulated to shape the structure and function of some microbial communities, such as those harboured in the human gut[44]. The specific role vitamin exchange plays in shaping anammox community interactions currently remains unknown, and should therefore be investigated further in future studies.

The oxidization of amino acids coupled to the reduction of nitrate to nitrite by CHB1 would serve to enhance overall nitrogen removal in the ecosystem through the completion of a nitrite loop with *Brocadia sp.*, as proposed by Speth *et al.*[16] Amino acid catabolism by CHB1 would also result in the release of ammonia, allowing it to be removed from the system by *Brocadia sp.*, thus further enhancing overall nitrogen removal. It is also possible that nitrate reduction to nitrite by CHB1 and other heterotrophic organisms helps stimulate anammox metabolism. Recent microsensor studies have shown that nitrite concentrations can become limiting in the interior of anammox granules[45]. Therefore, the ability of CHB1 to rapidly recycle nitrate back to nitrite may be critical for supporting anammox growth, which is required for their own proliferation.

In addition to a nitrite loop, heterotrophic bacteria in anammox ecosystems may facilitate complete denitrification, either alone or based on metabolite exchange of denitrification intermediates with other organisms. Indeed, the segregation of denitrification intermediates across different organisms has been shown to reduce the accumulation of growth-inhibiting metabolites (for example, nitrite) and potentially accelerate denitrification rates[46]. The close spatial association of organisms in anammox granules may encourage such a distributed metabolic network[47,48]. While complete denitrification would also serve to remove nitrogen from the ecosystem, it may potentially be detrimental to anammox bacterial growth because it would consume nitrite, rather than make it available for *Brocadia sp.* Specifically, the heterotrophic organisms capable of using nitrite as an electron acceptor (CHB2, CFX1, CFX2, PRO1 and PRO2) via nitrite reductase (NirK/S or NrfHA) may compete with *Brocadia sp.*, similar to NOB in single-stage partial nitritation-anammox bioreactors.

Aside from proteinaceous substrates, many organisms in the anammox community were capable of using other electron donors for energy conservation, including acetate, H$_2$, formate or simple organic compounds. It is possible that these substrates are produced during fermentation reactions and/or the breakdown of polysaccharide compounds present in the EPS. Chlorobi organisms related to CHB2 may be the biggest contributor to this process, based on expression of PFOR, hydrogenases and ADP-forming ACS involved in H$_2$ and acetate production. The availability of such substrates may contribute to niche differentiation by heterotrophs in the community, while also providing additional reducing equivalents to fuel anammox metabolism[49]. Nevertheless, oxidative phosphorylation coupled to nitrate respiration, rather than fermentation, was found to be the main energy metabolism expressed across the heterotrophic genomes, consistent with the high availability of nitrate and nitrite in the ecosystem. As such, the significance of H$_2$ and acetate metabolism in anammox granules still remains to be determined.

In conclusion, our combined approach of genome-centric metagenomics with metatranscriptomics allowed us to obtain a much deeper understanding of ecosystem function in anammox granules that has broad implications for anammox-based wastewater treatment. This study provides a robust analysis of the metabolic networks underlying several poorly characterized taxa frequently detected in anammox granules (Planctomycetes, Chlorobi, Bacteroidetes, Chloroflexi and Proteobacteria) and also sheds light on their potential ecological roles and interactions. Our results implicate members of the Chlorobi affiliated with CHB1 that are broadly found in anammox granules as highly active protein degraders, possibly liberating amino acids bound in the EPS matrix for themselves and other organisms in the community. These amino acids likely originate from anammox bacteria, which also provide nitrate and potentially essential B-vitamins to the community, supporting anaerobic respiration and growth. Other heterotrophic community members also appear to contribute to the scavenging of detritus and peptides produced by anammox bacteria, and may potentially use alternative electron donors, such as H$_2$, acetate and formate. These substrates may be made available through the hydrolysis and fermentation of EPS and detritus.

While further studies are still required to determine the specific niche space occupied by members of the community, the metabolic characterization provided here advances our insight on the ecological roles played by both anammox and heterotrophic bacteria in anammox granules. The high expression of enigmatic functions by the community, including hypothetical proteins and HAO-like proteins, together with our incomplete understanding of the biochemical pathways underlying EPS biosynthesis, highlights the need for continued work on these novel and industrially important ecosystems.

## Methods

**Bioreactor operation.** A 5 l continuously fed sequencing batch reactor (anammox bioreactor) has been operating in the Goel Laboratory at the University of Utah for the past 6 years, achieving stable anammox performance[50]. The anammox bioreactor was originally inoculated with anammox biomass from the City College of New York (Civil Engineering Department) that was enriched from activated sludge. The bioreactor was fed with anaerobic digester filtrate obtained from the belt filter press of a local wastewater treatment plant (CVWRF, Salt Lake City, UT, USA). Before entering the anammox reactor, the filtrate was first passed through a nitritation reactor, where approximately half of the influent ammonia was oxidized to nitrite. The nitritation reactor had a working volume of 2 l with a hydraulic retention time of 1 day[50]. Dissolved oxygen in the reactor was maintained at $0.5 \pm 0.5 \, mg \, l^{-1}$ and bicarbonate present in the influent filtrate kept the pH buffered between 6.5 and 8.5. The anammox bioreactor was operated at room temperature with a hydraulic retention time of 2 days and no biomass wasting, which encouraged the formation of dense granules. The pH of the anammox bioreactor was maintained at between 6.8 and 8.0 and the reactor contents were completely mixed using a stir bar and gas sparging. Anaerobic conditions in the anammox bioreactor were maintained by continuously purging a mixture of 95% nitrogen gas and 5% carbon dioxide[50]. Table 2 summarizes the process performance data of the nitritation and anammox bioreactors during steady-state operation.

**DNA and RNA sequencing.** Two independent biomass samples were collected from the anammox bioreactor at different time points during periods of high total nitrogen removal efficiency (>80%). DNA was extracted separately from each biomass sample to improve the recovery of MAGs and total RNA was extracted from the second biomass sample to examine steady-state community gene expression. 500 mg of biomass was collected from the reactor, centrifuged at 4500 r.p.m. for 5 min at 4 °C to remove the supernatant, and instantly flash frozen in liquid nitrogen. Total genomic DNA and RNA was extracted from biomass pellets using the PowerMax Soil DNA Isolation Kit (MoBio Laboratories, USA) and the PureLink RNA mini kit (Life Technology, NY, USA), respectively, according to the manufacturers' protocols. Genomic DNA was quality-checked using agarose gel electrophoresis and a Nanodrop ND-2000c (Thermo Fisher Scientific, USA). Following RNA extraction, residual genomic DNA was removed from total RNA using an on-column PureLink DNase set (Life Technologies, NY, USA). Total RNA quality and quantity was subsequently checked using the Bioanalyzer RNA 6000 Nano Assay (Agilent, Santa Clara, CA, USA) to ensure only high-quality nucleic acids were used for downstream analysis. Total RNA of 100 ng was used to construct strand specific RNA-Seq libraries with the Encore Complete Prokaryotic RNA-Seq DR Multiplex System (NuGEN, San Carlos, CA, USA). Non-rRNA in RNA-Seq libraries were enriched by selective priming during the first strand cDNA synthesis reaction, as well as in the final library construction steps using manufacturer's protocols. DNA from the first sample was sequenced on the Illumina MiSeq platform (Illumina, CA, USA) to generate 300 bp paired-end reads (550 bp mean insert size). DNA and RNA from the second sample were sequenced on the Illumina HiSeq 2000 platform to generate 125 bp paired-end reads (180 bp mean insert size) at greater sequencing depth. All DNA and RNA sequencing was performed at the Huntsman Cancer Institute (HCI), University of Utah. Raw DNA and RNA sequences can be found on the National Center for Biotechnology Information (NCBI) website under BioProject PRJNA343219.

**Metagenomic assembly and binning.** Raw paired-end reads from the MiSeq and HiSeq platforms were initially filtered using Sickle v1.33 (ref. 51) based on a minimum quality score of 20, a minimum sequence length of 100 bp and allowing for no ambiguous bases. Paired-end reads were then merged using FLASH v1.2.11 (ref. 52) and co-assembled using the de novo assembler of CLC Genomics Workbench v7.0.3 (CLCbio, Arhus, Denmark) based on default parameters (word size = 20 and bubble size = 50). The per-base coverage depth across all contigs was calculated by mapping raw reads from each sample against the co-assembled contigs using BBMap v35.92 (https://sourceforge.net/projects/bbmap/) with the parameters 'minid = 0.95' and 'ambig = random'. Resulting mapping files were subsequently used by MetaBAT v0.26.3 to bin metagenomic contigs into draft genomes, based on the 'sensitive' parameters[53]. CheckM v1.0.3 (ref. 54) was used to estimate the contamination and completeness of each draft genome based on 111 essential single-copy marker genes[55]. Draft genomes have been deposited to GenBank under the accession numbers provided in Supplementary Table 3.

**Phylogenetic analysis of recovered draft genomes.** Phylogenetic analysis of the recovered draft genomes was accomplished using Phylosift v1.0.1, based on a set of 37 universal single-copy marker genes[56]. The taxonomic affiliation of each draft genome was determined using the Phylosift 'all' command. Marker genes were also identified in 33 publicly available genomes closely related to the recovered draft genomes and used to build a phylogenetic tree. Marker genes were concatenated and aligned using Phylosift and a maximum likelihood tree was generated using RAxML v8.2.4 with the automatic protein model assignment algorithm (PROTGAMMAAUTO) and 100 bootstraps.

**Genome annotation and metabolic reconstruction.** Metabolic reconstruction of the recovered draft genomes was performed using MetaPathways v2.5 (ref. 20). Briefly, ORFs were predicted using Prodigal v2.0 (ref. 57), based on a minimum nucleotide length of 60, and queried against the SEED subsystems (accessed March 2013), Clusters of Orthologous Groups (COG, accessed December 2013), RefSeq (accessed September 2015) and MetaCyc (accessed October 2011) protein databases using the optimized LAST algorithm (E value, 1E-6) for functional annotation[58]. Contig nucleotide sequences were also queried against the SILVA small subunit (SSU) 123 database to identify the taxonomy of recovered 16S rRNA genes. Annotated genomes were then used to reconstruct the metabolic network of each organism using Pathway Tools and the MetaCyc database[59]. Pathway/Genome Databases were created for each genome, where pathway inference was based on a set of rules used by the Pathway Tools prediction algorithm Pathologic, including the presence of all key reactions and the completeness of the reconstructed pathway[60]. All inferred pathways were then manually curated to verify predictions made by Pathologic.

**Metatranscriptomic analysis.** Metatranscriptomic reads were quality filtered and merged as described above. Subsequently, rRNA sequences were filtered from the metatranscriptomic data set using SortMeRNA v2.0, based on multiple rRNA databases for bacterial, archaeal and eukaryotic sequences[61]. Resulting non-rRNA reads were mapped back to all assembled metagenomic contigs using BBMap v35.92 with the parameters 'minid = 0.95', which specifies a minimum alignment identify of 95% corresponding to the well-established criteria for identifying microbial species using average nucleotide identify[62] and 'ambig = random', which ensures reads with multiple top-scoring mapping locations are assigned randomly to a single location. Read counts were calculated for each predicted ORF using htseq-count v0.6.1 with the 'intersection strict' parameter[63] and normalized by sequencing depth and ORF length and expressed as RPKM values[64]. This allowed for the profiling of gene expression across the recovered draft genomes. Relative gene expression within each genome was calculated by relativizing the expression (RPKM value) of each ORF by the median RPKM value calculated across the genome. Pathway expression levels were calculated based on averaging the RPKM values for each reaction in a given pathway. For reactions with multiple genes, the highest expressed gene between multi-copy genes was selected, whereas the lowest expressed gene between multi-gene enzyme complexes (for example, hydrazine synthase, *hzsABC*) was selected.

The relative abundance and gene expression of the recovered genomes was calculated from the total number of DNA or mRNA reads that mapped to the genome, divided by the genome length (read count/genome size).

**Carbohydrate hydrolase and peptidase identification.** Carbohydrate hydrolases and peptidases potentially involved in the breakdown of EPS were identified in each genome based on BLASTP searches against the CAZy (accessed on September 2014)[65] and MEROPS release 10.0 (ref. 66) databases, respectively. Manual searches were also performed across all genomes based on gene annotations, and protein domains with hydrolytic activity were subsequently confirmed based on queries against the NCBI's Conserved Domain Database[67] and Pfam database[68] using MOTIF (http://www.genome.jp/tools/motif/). The subcellular location of identified proteins was predicted using CELLO v2.5, which classifies proteins using support vector machines trained by multiple feature vectors based on *n*-peptide composition[69].

**Data availability statement.** All data associated with this project can be found at the NCBI under BioProject PRJNA343219. Illumina HiSeq metagenomic data can be found under BioSample SAMN05785375, Illumina MiSeq metagenomic data can be found under BioSample SAMN05785373, and Illumina HiSeq metatranscriptomic data can be found under BioSample SAMN05785376. Annotated GenBank files for the whole genome sequences (MAGs) described in this study can be found under accession numbers listed in Supplementary Table 3.

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

## Acknowledgements

This work was supported by funding from the Central Valley Water Reclamation Facility (CVWRF) and the North Davis Sewer District awarded to R.G. and S.W.; Partial support was also provided by funding from the National Science Foundation (CBET-1435661 and MCB-1518130) awarded to D.R.N. and K.D.M., and a Postgraduate Scholarship-Doctoral (PGS-D) awarded by the National Sciences and Engineering Research Council of Canada (NSERC) to C.E.L.; The authors acknowledge Masaru K. Nobu for helpful discussion with the manuscript.

## Author contributions

C.E.L., K.D.M., R.G. and D.R.N. designed the study. S.W. and A.S.B. operated the bioreactor and performed the sampling and sequencing. C.E.L. analysed the data and wrote the manuscript. J.J.H. and A.S.B. contributed to the data analysis. J.J.H., A.S.B., K.D.M., R.G. and D.R.N. reviewed and provided valuable edits to the manuscript.

## Additional information

**Competing interests:** The authors declare no competing financial interests.

