## [Peer Review File · Nature Communications]

Reviewers' Comments:

Reviewer #1 (Remarks to the Author)

This manuscript describes roles of each member within a microbial community in the anammox granules. The approach employed here, namely combining metagenomics, genome-binning and metatranscriptomics, is the appropriate way for describing the role of community members. Especially, over 80% of mapping rate of raw reads (both DNA and RNA) indicates that the sequence data generated here is high enough to describe the function of the entire community. In addition, high abundance of anammox member within the granules and also high expression of genes related to the anammox reaction demonstrate that this reactor has been operated properly. Therefore, even though the result has been concluded from one snap-shot dataset in a single lab reactor, the knowledge delivered from this study must include a true view of the anammox community, thus this manuscript must be important for this field of scientist and also must contribute the further system development. Meanwhile, the community is quite simple compared with Speth (2016), and this is probably because this study uses a lab-reactor, not a full scale reactor. However, this reviewer sees that the metatranscriptomics data contributes to the deeper understanding of anammox cooperative relationship, so this study is novel enough to be published in the journal.

I have a couple of concerns in the sequence data analysis and also the presentation way.

- 1) Relative gene expression is used to evaluate the gene expression level. However, to show the gene expression level, principally, value of cDNA/DNA ratio (RNA RPKM/DNA RPKM) for each gene needs to be employed because expression level of duplicated or triplicated genes (orthologues) or genes including specific repeats should be overestimated if the relative gene expression is evaluated by the relative RNA RPKM ($\sim \text{RNA RPKM} / \text{Average coverage of the genome}$) value.
- 2) General understanding is that anammox is one of the most difficult energy metabolic reaction. However, the anammox provides energy (carbons/amino acids) to the other heterotrophs, which is interesting. The fact indicates that the relationship is energetically favorable under this condition and metatranscriptomics must contribute to illustrate the details of this cooperative relationships. Related to this, I have a couple of concerns, questions and comments to this manuscript:

- To describe the roles of each organism in a community based on the omics data, the environmental chemistry data at the sampling point for the omics studies needs to be shown somewhere in the paper. In this case, for example, the readers probably want to know TOC, DOC (if present), total inorganic carbon, and concentrations of NH_4 , NO_3 , NO_2 , and nitrogen removal rate etc..., even if the referred paper describes details of reactor operation. Regarding to this, it would be kind to readers if the concentration of NH_4 , NO_3 and NO_2 is incorporated into Figure 6. Figure 6 is just focused on presenting interaction between the two organisms, and ignore external input to this granule except for CO_2 . But of course, without the external input, the nitrate loop cannot be completed. So that should be shown in the figure.

-- I recommend to move Figure S1 to the main text with some modifications. In my understanding, nobody has presented about the fact based on the community gene expression data.

-- Order of figures/story may want to be reorganized: Figure 1, Figure S1 and Figure 4 are about the relative abundance of members or the relative expression level of the genes within the entire community while Figure 3 & 5 are about the relative expression level of the gene (set) within the individual organisms. The meaning of the values relative to the community or to the individual organism is totally different, but since these two are presented alternately, this could mislead the readers. Thus, I recommend to present Figure 1, S1 and 4 first and describe about the findings associated with the N removal in the entire system (community), and then to present Figure 3 & 5 about the function of individual organisms. In addition, it should be interesting if you prepare the figure 3 type of data but using the values relative to the entire community (RNA RPKM). That can show the contributing members for a given function in the entire system. In addition, to me, is quite reasonable/interesting that CHB2 lost the capability to convert NO_2^- to NO .

-- Based on the gene expression level of CHB2 (shown in Table S3), the major life style must be respiration, not fermentation. So my understanding from figure 6 and Table S3 is that the CHB2 uses NO_3^- (or perhaps N_2O) as the electron acceptor and the potential electron donor is amino acid. However, the latter part of conclusion, i.e. electron donor side, is not convincible only from the presented data in this main text (Fig. 5). I don't see much differences of the relative gene expression between amino acid synthesis vs degradation for CHB2. If the authors want to conclude that CHB2 is an active degrader of extracellular protein, different metatranscriptomics data needs to be presented in the main text. Probably like extracellular proteinase and amino acid transporter. (e.g. Table S6), and also may want to include how the CHB2 relates to the other potential electron donors, like sugar, organic acids etc... (I am not sure CHB3 encode such genes though.)

- Figure 3: What is the yellowish color box? No pathways in the genome?

Reviewer #2 (Remarks to the Author)

This study utilized genome-centric metagenomics and metatranscriptomics to reveal the metabolic capability and interactions among anammox and heterotrophic bacteria within a laboratory-scale anammox bacteria. The authors were able to utilize DNA sequences from two different samples and sequencing runs to perform metagenomic binning. Metatranscriptomics was only performed on one biological sample and thus gene expression values were given relative to the average RPKM value of all genes across a binned genome. While genome-centric metagenomics has been applied to study anammox granules previously, this is the first time an approach has been combined it with metatranscriptomics. Through this approach, the authors identify some heterotrophs in the granule as playing a role in degrading EPS, the role of heterotrophs in the nitrite loop, and a potential symbiotic relationship between heterotrophs and anammox bacteria related to B-vitamins.

While the manuscript is generally well-written, the authors use some terminology throughout that could be misleading and thus needs to be revised. For instance, the term "activity" is used alone without qualifying if the authors mean "gene expression" activity or "enzyme" activity. In addition, the authors need to be careful that they are talking about the expression of genes and not enzymes throughout the Results and Discussion section. These are important distinctions because gene expression activity rarely equates or even correlates to enzyme activity.

Specific Comments

Abstract

Page 2, Line 7 - Care should be taken when using the term "activity", as it is typically associated with enzyme activity. In this paper, it seems that the authors are examining the actively expressed genes within different binned genomes of anammox and heterotrophic, in order to identify their potential interactions.

Methods

Page 6, Line 1 - Although the nitrification reactor was not the focus of this manuscript, more detail on the design/operation of the nitrification reactor should be given.

Results

Page 11, Lines 11, 12, and 18-20 - Again it is suggested the term "activity" be changed to "gene

expression"

It needs to be more clear that the authors are talking about the expression of genes and not proteins. Many sentences could be easily edited by adding the word "genes" to sentences (i.e., "the genes encoding for XXXX were expressed" or "XXXX genes were expressed"). For instance, the sentence on Page 14, Lines 1-3 could be changed to "All heterotrophs expressed respiratory nitrate reductase genes (narGHIJ), which requires the transport of nitrate into the cytoplasm. Other places include (but are not limited to):

Page 14, Lines 1-12

Page 15, Line 7

Page 16, Line 18

Page 17, Lines 4-8

Page 20, Line 13

Page 22, Line 6

Page 19, Lines 4-5 - the end of the sentence is worded awkwardly and should be changed to "...involved in the biosynthesis pathways of B-vitamins.

Page 19, Lines 11-12 - This is an interesting hypothesis that suggests the formation of symbiotic relationships within the granules and it is suggested that this be expanded upon in the Discussion section. In particular, the authors should explain if transporters for B-vitamins were found in the genomes of CHB2, CHB7 and CHX17. Figure 6 does not show a specific transport protein B-vitamin transport for CHB2, so does this mean were they not identified? Also, it is mentioned that key genes in different B-vitamin de novo synthesis pathways are missing, is it possible from the data given that the loss of the genes was a recent evolutionary event? The authors should also mention why these B vitamins might be important to these heterotrophic bacteria (for instance, their potential role in the breakdown of polysaccharides).

Discussion

Page 19, Line 16 - Again, it is suggested that the authors be cautious of the use of "metabolic activity", as the analyses performed relate more to verifying and identifying the metabolic role/function than actually measuring activity.

Figures and Tables

Page 21, Line 2 - Change "activity" relative transcript abundance

Page 29, Line 11-14 - It needs to be clearly stated in the caption that the values are "relative".

Reviewer #3 (Remarks to the Author)

in their manuscript entitled 'Metabolic network analysis reveals microbial community interactions in anammox granules' Lawson et al describe the metagenomic and metatranscriptomic sequencing of a lab-scale granular anammox reactor. The studied reactor is part of a 2-step nitrification anammox system used in wastewater treatment, is and fed with filtered wastewater. The analyses focus on the roles of the non-anammox community members within the granule. As the system is moderate complexity, the sequencing effort is sufficient to capture the majority of the diversity in the reactor. This enables the authors to evaluate the community as a whole, rather than just the genomes of several members.

This work is timely, as anammox-based nitrogen removal from wastewater is increasingly adopted, and understanding the microbial community in treatment systems is key to further improve the effectiveness of these systems. As the authors point out, there is some overlap with the work of Speth et al. published earlier this year. However, the differences between the systems (1-step vs 2-step) and addition of transcription data in my opinion give this work substantial added value.

I thus recommend 'Metabolic network analysis reveals microbial community interactions in anammox granules' for publication, provided that the authors address the remarks/questions below.

page 6 line 20: "quality-checked"

page 7 lines 7-9: please specify the sequenced fragment length here. Table S1 suggests overlapping pairs by a minimum of 10bp

page 7 line 19: Please specify the used assembly parameters here (at least word/bubble size). CLC stores previously used parameters, so default equals 'last-used'.

page 8 line 21: please clarify if 'Nucleotide sequences' refers to assembled contigs or reads

page 9 line 15: RPKM is 'reads per kilobase of exon model per million mapped reads' please change 'reads per kilobase million' to be a little more explicit.

page 9 line 20: What do the authors mean with "reactions with multiple genes"? if multiple copies of the same gene, using the highest expressed one is fine. If it refers to a multi-gene complex (eg HzsABC) I do not agree with the approach of picking the highest expressed gene in

the complex. As the lowest expressed subunit might be limiting that seems more appropriate.

page 10 lines 15-16: N50 and N20 are meaningless parameters in a metagenome context. I suggest the authors remove them from the text.

page 10 line 21: please also specify the fraction of mRNA reads that could be mapped to the draft genomes.

page 11 line 12: It is unclear to me how "relative abundance" and "relative expression" are calculated. Personally, I'd use overall nucleotide coverage (nucleotides mapped / reference nucleotides) for both abundance and expression here. Additionally, table S1 seems to imply that only the reads from the 2nd sampling were used in this analysis (which I agree is the correct approach). If so, please specify this in the text.

Page 11 line 22: if this high expression levels are reason to include these genomes in the subsequent analyses, please specify which ones.

page 13 lines 7-8: *Brocadia fulgida* does in fact contain a *nirK* like sequence that closely resembles the *nirK* of KSU-1 (*Jettinia asiatica*). The correct citation for the *Brocadia fulgida* genome is Ferousi et al 2013 "Identification of the type II cytochrome c maturation pathway in anammox bacteria by comparative genomics". Gori et al only assessed the presence of a set of known anammox genes in an unassembled metagenome and did not look for *nirK*.

page 13 lines 10-16: The metabolism of nitrite in anammox bacteria is an unresolved question. I suggest the authors explicitly mention both possibilities, that one of the HAOs could perform the reduction of NO₂ to either NO or NH₂OH.

page 14 line 3: reductase

page 15 line 3-4: I suggest to remove 'encoded pathways for CO₂ fixation.' and merge sentences

page 17 line 1: 'alternative electron donors' alternative to what?

page 17 line 2: "other simple organic" implies all donors mentioned are organics

page 20 line 4: Speth

page 21: it is an open question in anammox research whether the community provides any metabolites to anammox bacteria. Using the CHB2 genome + expression data it would be relatively easy to check if there are any suspiciously expressed biosynthetic gene clusters (as

predicted by eg antiSMASH) in this genome.

page 21 line 9-11: breakdown of amino acids coupled to nitrate respiration would not only release NO₂ but also NH₄, possibly even further enhancing the N removal in the system.

page 29/30/31: change respectfully to respectively

Figure 3 - Legend

I assume the beige color in figure 3 indicates pathway absence. Please state this in the legend. Additionally, please specify that a value of 1 reflects median expression 'in a given genome'

Figure 3 - reversible pathways

How do the authors distinguish between reversible pathways in fig 3? In the case of hydrogen production/oxidation it looks like the expression is identical and the lines should be merged. However, in the case of glycolysis/gluconeogenesis BCD10 and CFX17 show differential expression

figure 3 - respiration

this category seems to imply that the other processes are not respiration? Also, can the authors clarify what 'aerobic respiration (cytochrome c)' includes

Figure 3 & 4

In my opinion, figure 4 doesn't add much to the information present in figure 3. I recommend splitting the 'denitrification' line in figure 3 into NO₃ -> NO₂ & NO₂ -> N₂ and removing figure 4 altogether. Additionally, it seems that the expression levels of fig 3 and fig 4 don't agree, especially in case of CHB2 and PRO8.

Figure 5 - absence of genes

Unlike figure 3, there seems to be no distinction between pathway absence and (very) low expression. Please indicate pathway absence in figure 5 as well.

Figure 6

The figure only shows Trp as incorporated in protein synthesis in CHB2. I'd argue that all AA that CHB2 can't synthesize & break down are used this way. Additionally, the figure ignores the presence of a periplasm. If this was done for clarity of the schematic, please state this in the legend

Table 1 - Bin IDs

I assume the bin ids reflect the numbers generated by metabat, but I suggest the authors renumber them for use in the manuscript

Table 1 - AMX1 genome completeness

The genome of AMX1 is just over 3Mb in size, substantially smaller than previously published anammox genomes. This raises some concern with regard to genome completeness. If there is strain-level variation present, variable (pan-genomic) regions of the genome will not be binned correctly as their coverage will be lower (reads originate from 1 strain vs 2 strain) in the assembly. As such variable regions don't contain single copy marker genes, the genome will still show up as near-complete using tools such as checkM. I recommend the authors assess strain level variation qualitatively by checking the contigs for SNV's in CLC and reassess bin completeness if strain level variation is present.

Reviewers' comments:

Response to reviewers' in blue.

Reviewer #1 (Remarks to the Author):

This manuscript describes roles of each member within a microbial community in the anammox granules. The approach employed here, namely combining metagenomics, genome-binning and metatranscriptomics, is the appropriate way for describing the role of community members. Especially, over 80% of mapping rate of raw reads (both DNA and RNA) indicates that the sequence data generated here is high enough to describe the function of the entire community. In addition, high abundance of anammox member within the granules and also high expression of genes related to the anammox reaction demonstrate that this reactor has been operated properly. Therefore, even though the result has been concluded from one snap-shot dataset in a single lab reactor, the knowledge delivered from this study must include a true view of the anammox community, thus this manuscript must be important for this field of scientist and also must contribute the further system development. Meanwhile, the community is quite simple compared with Speth (2016), and this is probably because this study uses a lab-reactor, not a full scale reactor. However, this reviewer sees that the metatranscriptomics data contributes to the deeper understanding of anammox cooperative relationship, so this study is novel enough to be published in the journal.

I have a couple of concerns in the sequence data analysis and also the presentation way.

1) Relative gene expression is used to evaluate the gene expression level. However, to show the gene expression level, principally, value of cDNA/DNA ratio (RNA RPKM/DNA RPKM) for each gene needs to be employed because expression level of duplicated or triplicated genes (orthologues) or genes including specific repeats should be overestimated if the relative gene expression is evaluated by the relative RNA RPKM ($\sim \text{RNA RPKM} / \text{Average coverage of the genome}$) value.

Response:

The goal of our metatranscriptomic analysis was to determine which ORFs in a given genome had the highest overall expression (i.e. RNA RPKM), as this corresponds with potential enzyme activity. Relativizing the ORF RNA RPKM values to the genome median RNA RPKM therefore makes sense for calculating relative expression, provided that orthologous genes or repeat regions are not counted multiple times, as the reviewer points out.

To alleviate the reviewers concerns, we have redone all mapping analysis described in this paper using BMap (<http://jgi.doe.gov/data-and->

tools/bbtools/), which ensures reads with multiple top-scoring mapping locations (e.g. orthologs or repeat regions) are assigned randomly to a single location, thus preventing overestimation of gene expression values (see revised Methods, page 9, lines 17-21). This was not accounted for in our initial analysis with BWA-mem and we thank the reviewer for pointing out this potential source of error. The changes in mapping software had only minor impact on the results and did not change any conclusions from the study.

2) General understanding is that anammox is one of the most difficult energy metabolic reaction. However, the anammox provides energy (carbons/amino acids) to the other heterotrophs, which is interesting. The fact indicates that the relationship is energetically favorable under this condition and metatranscriptomics must contribute to illustrate the details of this cooperative relationships. Related to this, I have a couple of concerns, questions and comments to this manuscript:

- To describe the roles of each organism in a community based on the omics data, the environmental chemistry data at the sampling point for the omics studies needs to be shown somewhere in the paper. In this case, for example, the readers probably want to know TOC, DOC (if present), total inorganic carbon, and concentrations of NH₄, NO₃, NO₂, and nitrogen removal rate etc..., even if the referred paper describes details of reactor operation. Regarding to this, it would be kind to readers if the concentration of NH₄, NO₃ and NO₂ is incorporated into Figure 6. Figure 6 is just focused on presenting interaction between the two organisms, and ignore external input to this granule except for CO₂. But of course, without the external input, the nitrate loop cannot be completed. So that should be shown in the figure.

We have included a new table containing the process performance and kinetic data from both reactors (Table 1). We better highlighted external inputs to the system in Figure 6.

-- I recommend to move Figure S1 to the main text with some modifications. In my understanding, nobody has presented about the fact based on the community gene expression data.

Genes that may catalyze nitrite reduction to hydroxylamine in *Brocadia sinica* have been identified by Oshiki et al (PMID:27112128). We have identified orthologous genes in our *Brocadia* sp. genome and included expression information for them (Fig S1). Given the number of figures in the main text (7 total), we feel that this should still remain in the supplementary materials.

-- Order of figures/story may want to be reorganized: Figure 1, Figure S1 and Figure 4 are about the relative abundance of members or the relative expression

level of the genes within the entire community while Figure 3 & 5 are about the relative expression level of the gene (set) within the individual organisms. The meaning of the values relative to the community or to the individual organism is totally different, but since these two are presented alternately, this could mislead the readers. Thus, I recommend to present Figure 1, S1 and 4 first and describe about the findings associated with the N removal in the entire system (community), and then to present Figure 3 & 5 about the function of individual organisms. In addition, it should be interesting if you prepare the figure 3 type of data but using the values relative to the entire community (RNA RPKM). That can show the contributing members for a given function in the entire system. In addition, to me, is quite reasonable/interesting that CHB2 lost the capability to convert NO₂⁻ to NO.

We have reorganized the figures and the text based on the above recommendations. Figures 1, 2, and 4 are now presented first, followed by Figures 3, 5, 6, 7.

-- Based on the gene expression level of CHB2 (shown in Table S3), the major life style must be respiration, not fermentation. So my understanding from figure 6 and Table S3 is that the CHB2 uses NO₃ (or perhaps N₂O) as the electron acceptor and the potential electron donor is amino acid. However, the latter part of conclusion, i.e. electron donor side, is not convincible only from the presented data in this main text (Fig. 5). I don't see much differences of the relative gene expression between amino acid synthesis vs degradation for CHB2. If the authors want to conclude that CHB2 is an active degrader of extracellular protein, different metatranscriptomics data needs to be presented in the main text. Probably like extracellular proteinase and amino acid transporter (e.g. Table S6), and also may want to include how the CHB2 relates to the other potential electron donors, like sugar, organic acids etc... (I am not sure CHB3 encode such genes though.)

In response to the reviewer suggestion, we have created a new figure showing peptidase and transporter expression (Figure 6 in revised manuscript).

- Figure 3: What is the yellowish color box? No pathways in the genome?

Yes, yellow box indicates pathway absence. We have added the explanation to the figure legend (Figure 4 in revised manuscript).

Reviewer #2 (Remarks to the Author):

This study utilized genome-centric metagenomics and metatranscriptomics to reveal the metabolic capability and interactions among anammox and heterotrophic bacteria within a laboratory-scale anammox bacteria. The authors were able to utilize DNA sequences from two different samples and sequencing

runs to perform metagenomic binning. Metatranscriptomics was only performed on one biological sample and thus gene expression values were given relative to the average RPKM value of all genes across a binned genome. While genome-centric metagenomics has been applied to study anammox granules previously, this is the first time an approach has been combined with metatranscriptomics. Through this approach, the authors identify some heterotrophs in the granule as playing a role in degrading EPS, the role of heterotrophs in the nitrite loop, and a potential symbiotic relationship between heterotrophs and anammox bacteria related to B-vitamins.

While the manuscript is generally well-written, the authors use some terminology throughout that could be misleading and thus needs to be revised. For instance, the term "activity" is used alone without qualifying if the authors mean "gene expression activity" or "enzyme activity". In addition, the authors need to be careful that they are talking about the expression of genes and not enzymes throughout the Results and Discussion section. These are important distinctions because gene expression activity rarely equates or even correlates to enzyme activity.

Thanks for the comment. We have clarified that we are referring to gene expression activity throughout the manuscript.

Specific Comments

Abstract

Page 2, Line 7 - Care should be taken when using the term "activity", as it is typically associated with enzyme activity. In this paper, it seems that the authors are examining the actively expressed genes within different binned genomes of anammox and heterotrophic, in order to identify their potential interactions.

We have changed the term "activity" to "gene expression activity"

Methods

Page 6, Line 1 - Although the nitritation reactor was not the focus of this manuscript, more detail on the design/operation of the nitration reactor should be given.

We have added more details regarding the nitritation reactor to the methods. We have also added a new table describing the process data (Table 1).

Results

Page 11, Lines 11, 12, and 18-20 - Again it is suggested the term "activity" be changed to "gene expression"

It needs to be more clear that the authors are talking about the expression of genes and not proteins. Many sentences could be easily edited by adding the word "genes" to sentences (i.e., "the genes encoding for XXXX were expressed" or "XXXX genes were expressed"). For instance, the sentence on Page 14, Lines 1-3 could be changed to "All heterotrophs expressed respiratory nitrate reductase genes (narGHIJ), which requires the transport of nitrate into the cytoplasm. Other places include (but are not limited to):

Page 14, Lines 1-12

Page 15, Line 7

Page 16, Line 18

Page 17, Lines 4-8

Page 20, Line 13

Page 22, Line 6

We agree with this comment and have edited all sentences to clarify we are talking about gene expression.

Page 19, Lines 4-5 - the end of the sentence is worded awkwardly and should be changed to "...involved in the biosynthesis pathways of B-vitamins.

We have revised the sentence for clarity.

Page 19, Lines 11-12 - This is an interesting hypothesis that suggests the formation of symbiotic relationships within the granules and it is suggested that this be expanded upon in the Discussion section. In particular, the authors should explain if transporters for B-vitamins were found in the genomes of CHB2, CHB7 and CHX17. Figure 6 does not show a specific transport protein B-vitamin transport for CHB2, so does this mean were they not identified? Also, it is mentioned that key genes in different B-vitamin de novo synthesis pathways are missing, is it possible from the data given that the loss of the genes was a recent evolutionary event? The authors should also mention why these B vitamins might be important to these heterotrophic bacteria (for instance, their potential role in the breakdown of polysaccharides).

In response to this comment, we further analyzed the data to identify B-vitamins transporters in the genomes and now include this information in the results section of the revised manuscript (Page 20, lines 10-15) and added some discussion (Page 22, lines 11-17).

Discussion

Page 19, Line 16 - Again, it is suggested that the authors be cautious of the use of "metabolic activity", as the analyses performed relate more to verifying and identifying the metabolic role/function than actually measuring activity.

We agree and we have revised the text.

Figures and Tables

Page 21, Line 2 - Change "activity" relative transcript abundance

Done.

Page 29, Line 11-14 - It is needs to be clearly stated in the caption that the values are "relative".

Added relative.

Reviewer #3 (Remarks to the Author):

in their manuscript entitled 'Metabolic network analysis reveals microbial community interactions in anammox granules' Lawson et al describe the metagenomic and metatranscriptomic sequencing of of a lab-scale granular anammox reactor. The studied reactor is part of a 2-step nitritation anammox system used in wastewater treatment, is and fed with filtered wastewater. The analyses focus on the roles of the non-anammox community members within the granule. As the system is moderate complexity, the sequencing effort is sufficient to capture the majority of the diversity in the reactor. This enables the authors to evaluate the community as a whole, rather than just the genomes of several members.

This work is timely, as anammox-based nitrogen removal from wastewater is increasingly adopted, and understanding the microbial community in treatment systems is key to further improve the effectiveness of these systems. As the authors point out, there is some overlap with the work of Speth et al. published earlier this year. However, the differences between the systems (1-step vs 2-step) and addition of transcription data in my opinion give this work substantial added value.

I thus recommend 'Metabolic network analysis reveals microbial community interactions in anammox granules' for publication, provided that the authors address the remarks/questions below.

page 6 line 20: "quality-checked"

Done.

page 7 lines 7-9: please specify the sequenced fragment length here. Table S1 suggests overlapping pairs by a minimum of 10bp

We have added the insert sizes used during library preparation to the methods.

page 7 line 19: Please specify the used assembly parameters here (at least word/bubble size). CLC stores previously used parameters, so default equals 'last-used'.

We have added the assembly parameters to the text (Word size = 20, bubble size =50)

page 8 line 21: please clarify if 'Nucleotide sequences' refers to assembled contigs or reads

We have clarified that contig nucleotide sequences were used.

page 9 line 15: RPKM is 'reads per kilobase of exon model per million mapped reads' please change 'reads per kilobase million' to be a little more explicit.

We have changed the text to read, “reads per kilobase per million mapped reads”.

page 9 line 20: What do the authors mean with "reactions with multiple genes"? if multiple copies of the same gene, using the highest expressed one is fine. If it refers to a multi-gene complex (eg HzsABC) I do not agree with the approach of picking the highest expressed gene in the complex. As the lowest expressed subunit might be limiting that seems more appropriate.

Good point. We have recalculated reaction gene expression values using the following rules:

“For reactions with multiple genes, the highest expressed gene among multi-copy genes was selected, whereas the lowest expressed gene among multi-gene enzyme complexes (e.g. hydrazine synthase, HzsABC) was selected”

This has been added to the methods section (page 10, lines 6-11) and all figures have been revised based on recalculated values. The recalculation did not change any conclusions from the paper.

page 10 lines 15-16: N50 and N20 are meaningless parameters in a metagenome context. I suggest the authors remove them from the text.

We find these statistics valuable for understanding the length distribution of assembled metagenomic contigs and have respectfully opted to keep this information in the manuscript.

page 10 line 21: please also specify the fraction of mRNA reads that could be mapped to the draft genomes.

We have added this information to the “Metagenomic sequencing and binning” section.

page 11 line 12: It is unclear to me how "relative abundance" and "relative expression" are calculated. Personally, I'd use overall nucleotide coverage (nucleotides mapped / reference nucleotides) for both abundance and expression here. Additionally, table S1 seems to imply that only the reads from the 2nd sampling were used in this analysis (which I agree is the correct approach). If so, please specify this in the text.

Yes, relative abundance and gene expression were calculated from the total number of DNA or mRNA reads that mapped to the genome divided by the genome length (read count / genome size). We have added this to the methods section under “metatranscriptomic analysis”.

We have clarified that mapping results presented in Table S1 are based on the second biomass sampling (page 11, line 2).

Page 11 line 22: if this high expression levels are reason to include these genomes in the subsequent analyses, please specify which ones.

Thanks for catching this. We now specify which genomes we are referring to in the text.

page 13 lines 7-8: *Brocadia fulgida* does in fact contain a nirK like sequence that closely resembles the nirK of KSU-1 (*Jettienia asiatica*). The correct citation for the *Brocadia fulgida* genome is Ferousi et al 2013 "Identification of the type II cytochrome c maturation pathway in anammox bacteria by comparative genomics". Gori et al only assessed the presence of a set of known anammox genes in an unassembled metagenome and did not look for nirK.

Thank you for pointing this out. We have updated the citation and sentence in the text.

page 13 lines 10-16: The metabolism of nitrite in anammox bacteria is an unresolved question. I suggest the authors explicitly mention both possibilities, that one of the HAOs could perform the reduction of NO₂ to either NO or NH₂OH.

We have indicated that the NO₂ reduction to either NO or NH₂OH is possible (Page 14, lines 4-5 in revised manuscript).

page 14 line 3: reductase

Corrected.

page 15 line 3-4: I suggest to remove 'encoded pathways for CO₂ fixation.' and merge sentences

Done.

page 17 line 1: 'alternative electron donors' alternative to what?

We have changed the word “alternative” to “other” and modified the sentence.

page 17 line 2: "other simple organic" implies all donors mentioned are organics

We have modified this sentence for clarity.

page 20 line 4: Speth

Corrected.

page 21: it is an open question in anammox research whether the community provides any metabolites to anammox bacteria. Using the CHB2 genome + expression data it would be relatively easy to check if there are any suspiciously expressed biosynthetic gene clusters (as predicted by eg antiSMASH) in this genome.

We had found a large NRPS gene possibly involved in siderophore biosynthesis (dhbF) using antiSMASH. However, we want to avoid over speculation and have decided not to discuss this in the manuscript.

page 21 line 9-11: breakdown of amino acids coupled to nitrate respiration would not only release NO₂ but also NH₄, possibly even further enhancing the N removal in the system.

Very true. We have added this to the discussion (Page 22, lines 14-16).

page 29/30/31: change respectfully to respectively

Done.

Figure 3 - Legend

I assume the beige color in figure 3 indicates pathway absence. Please state this in the legend. Additionally, please specify that a value of 1 reflects median expression 'in a given genome'

Done.

Figure 3 - reversible pathways

How do the authors distinguish between reversible pathways in fig 3? In the case of hydrogen production/oxidation it looks like the expression is identical and the lines should be merged. However, in the case of glycolysis/gluconeogenesis BCD10 and CFX17 show differential expression

Hydrogen production/oxidation is a reversible pathway and these rows have been merged. Glycolysis/gluconeogenesis is mediated by different key enzymes (e.g. fructose biphosphate [EC 3.1.3.11] and phosphoenolpyruvate synthetase [EC 2.7.9.2]) and therefore will have different expression.

figure 3 - respiration

this category seems to imply that the other processes are not respiration? Also, can the authors clarify what 'aerobic respiration (cytochrome c)' includes

This is the official MetaCyc pathway name. It includes all complexes involved in the electron transport chain - NADH oxidoreductase (complex I), Succinate-Q oxidoreductase (complex II), ubiquinol-cytochrome c oxidoreductase (complex III), and cytC oxidase (complex IV). We have pointed the reader to Table S5 in the figure legend, which lists the reactions of each pathway.

Figure 3 & 4

In my opinion, figure 4 doesn't add much to the information present in figure 3. I recommend splitting the 'denitrification' line in figure 3 into $\text{NO}_3 \rightarrow \text{NO}_2$ & $\text{NO}_2 \rightarrow \text{N}_2$ and removing figure 4 altogether. Additionally, it seems that the expression levels of fig 3 and fig 4 don't agree, especially in case of CHB2 and PRO8.

We find Fig 4 valuable because it presents the expression and presence/absence information of all denitrification reactions. Adding the lines $\text{NO}_3 > \text{NO}_2$ & $\text{NO}_2 > \text{N}_2$ would not be as informative. Fig 4 also shows overall expression rather than relative expression (Fig 3), which allows the reader to compare expression levels across organisms (i.e. understand which organism might make the biggest overall contribution to denitrification in the reactor). Therefore, we would like to keep Fig 4. Note that these figures have switched order in the revised manuscript.

Figure 5 - absence of genes

Unlike figure 3, there seems to be no distinction between pathway absence and (very) low expression. Please indicate pathway absence in figure 5 as well.

We have included pathway absence on the figure (Now Fig 6 in the revised

manuscript).

Figure 6

The figure only shows Trp as incorporated in protein synthesis in CHB2. I'd argue that all AA that CHB2 can't synthesize & break down are used this way. Additionally, the figure ignores the presence of a periplasm. If this was done for clarity of the schematic, please state this in the legend

We agree with this statement and have revised Fig 6 to indicate that Trp is only an example of an AA that could be incorporated into protein synthesis (now Fig 7 in revised manuscript). We have also indicated that the periplasm has been left out for clarity in the figure legend.

Table 1 - Bin IDs

I assume the bin ids reflect the numbers generated by metabat, but I suggest the authors renumber them for use in the manuscript

We have renumbered the Bin IDs based on those submitted to NCBI.

Table 1 - AMX1 genome completeness

The genome of AMX1 is just over 3Mb in size, substantially smaller than previously published anammox genomes. This raises some concern with regard to genome completeness. If there is strain-level variation present, variable (pan-genomic) regions of the genome will not be binned correctly as their coverage will be lower (reads originate from 1 strain vs 2 strain) in the assembly. As such variable regions don't contain single copy marker genes, the genome will still show up as near-complete using tools such as checkM. I recommend the authors assess strain level variation qualitatively by checking the contigs for SNV's in CLC and reassess bin completeness if strain level variation is present.

This is an interesting and valid point brought up by the reviewer. While looking at SNV's in CLC might point to some strain-level diversity within the AMX1 population, it will not be able to explicitly determine whether a pool of contigs are missing from the genome because you will only be assessing those contigs belonging to the original bin. To identify both strain-level variation in the AMX1 genome, and also understand if specific genome regions of closely related *Brocadia* sp. are missing from AMX1, we competitively mapped back all raw metagenomic reads to AMX1 contigs, plus two genomes for *Brocadia sinica* (JZEK01 and BAFN01), at a cutoff of 60% nucleotide identity using BMap. If another abundant strain was present in the reactor, possibly convoluting coverage estimates of variable regions, we would expect to see significant reads mapping to the combined genomes at between ~70-85% (the ANI between our AMX1 bin and other genomes), representing AMX1 reads that are aligning to the missed regions present in the other genome (See Bendall et al, 2016, PMID: 26744812 for methods to determine population dynamics in mixed populations). The

figure below shows a histogram of the number of reads that mapped to these genomes at a given percent identify. As is observed from the figure, the majority of reads mapped to the AMX1 genome at >99% sequence identify, suggesting that strain heterogeneity among the AMX1 population was low. Moreover, if there was significant SNV's, you would expect the 98-99% ANI region (second bar from right) to be much greater.

We also note that the recently published genome of *Brocadia* sp. 40 (PMCID: PMC5146453) had a genome size of 2.93Mb, which is even smaller than our AMX1 genome. Ultimately, given that our AMX1 genome is a draft that is not closed, we cannot guarantee its size with certainty. However, based on the evidence provided, it seems unlikely that strain heterogeneity was prevalent in our AMX1 population.

Based on the above, we have decided to not change the discussion of the genome bins.

Reviewers' Comments:

Reviewer #1 (Remarks to the Author):

The revised manuscript is significantly improved. I don't see any concerns about this manuscript.

Reviewer #2 (Remarks to the Author):

The authors have addressed all reviewers' comments satisfactorily, making the manuscript acceptable for publication.

Reviewer #3 (Remarks to the Author):

In my opinion, the authors have addressed my remarks (and those of the other reviewers) in the rebuttal and the revised manuscript. I recommend the manuscript for publication.